# Two Generator Game: Learning to Sample via Linear Goodness-of-Fit Test

**Lizhong Ding**[1]   **Mengyang Yu**[1]   **Li Liu**[1]   **Fan Zhu**[1]
**Yong Liu**[2]   **Yu Li**[3]   **Ling Shao**[1]
[1]Inception Institute of Artificial Intelligence (IIAI), Abu Dhabi, UAE.
[2]Institute of Information Engineering, CAS, China.
[3]King Abdullah University of Science and Technology (KAUST), Saudi Arabia.

## Abstract

Learning the probability distribution of high-dimensional data is a challenging problem. To solve this problem, we formulate a deep energy adversarial network (DEAN), which casts the energy model learned from real data into an optimization of a goodness-of-fit (GOF) test statistic. DEAN can be interpreted as a GOF game between two generative networks, where one explicit generative network learns an energy-based distribution that fits the real data, and the other implicit generative network is trained by minimizing a GOF test statistic between the energy-based distribution and the generated data, such that the underlying distribution of the generated data is close to the energy-based distribution. We design a two-level alternative optimization procedure to train the explicit and implicit generative networks, such that the hyper-parameters can also be automatically learned. Experimental results show that DEAN achieves high quality generations compared to the state-of-the-art approaches.

## 1   Introduction

Learning the probability distribution of high-dimensional data, such as images and natural language corpora, is a challenging problem in machine learning. Traditionally, we define a parametric family of densities $\{p(x; \theta), \theta \in \Theta\}$ and find the one with the maximum likelihood using data $\{x_i\}_{i=1}^m$ in $\Theta$ (known as maximum likelihood estimation, MLE) [KW56]. However, the normalization factor introduces difficulties during the MLE training, because it is an integration over all configurations of random variables. Markov chain Monte Carlo (MCMC) [ADFDJ03, SEMFV17] could be used, but the distributions of real-world data, such as images, have an intriguing property, that probability mass is concentrated in sharp ridges that are separated by large low probability regions. This complexity of the probability landscape is a road block and a challenge that MCMC methods have to meet[BCV13].

Inspired by the representation ability of the hierarchical models of deep learning [HS06, HOT06, B+09, LBH15, GBCB16], and as an alternative to MLE approaches, generative adversarial networks (GANs) [GPAM+14] represent an important milestone on the path towards more effective generative models [RMC16, CDH+16, NCT16, ACB17b, ZML17, LCC+17, BSAG18]. GANs are a type of implicit generative models (IGMs), which generate images drawn from an unknown complex high-dimensional distribution $p(x)$, using an implicit distribution $q(x; \theta_g)$ usually represented by a deep network with parameter $\theta_g$. No estimation of likelihoods or exact inference are required in GAN-like models. This class of models has recently led to many impressive results [HLP+17, JZL+17, KALL17, BDS18], and different variants have been proposed for specific tasks, such as conditional GAN [MO14], Pix2Pix [IZZE17], CycleGAN [ZPIE17], starGAN [CCK+18] and etc.

In our opinion, the existing GAN models are fundamentally two-sample test problems. The goal of the two-sample test is to determine whether two distributions $p$ and $q$ are different, based on

samples $\mathcal{D}_x = \{x_i\}_{i=1}^n \subset \mathcal{X}$ and $\mathcal{D}_{x'} = \{x_j'\}_{j=1}^m \subset \mathcal{X}$ independently drawn from $p$ and $q$, respectively. GANs can be considered two-sample test problems because they need to decide whether the underlying distribution $p(x)$ of real data and an implicit distribution $q(x)$, which generates fake data, are different. From this perspective, we summarize existing GAN models into two categories based on how they measure the discrepancy between $p(x)$ and $q(x)$. The first category is the integral probability metric (IPM) [Mül97]. For a class $\mathcal{F}$ of functions, the IPM $\delta$ between two distributions $p$ and $q$ is defined as

$$\delta(p,q) = \sup_{f \in \mathcal{F}} \left| \int_{\mathcal{X}} f(x)p(x)dx - \int_{\mathcal{X}} f(x')q(x')dx' \right|.$$

If $\mathcal{F}$ is a class of Lipschitz functions, $\delta(p,q)$ is called the Wasserstein IPM. Wasserstein GANs (WGANs) were proposed based on the Wasserstein IPM [ACB17a, GAA+17]. If $\mathcal{F}$ is a unit ball within the reproducing kernel Hilbert space (RKHS), $\delta(p,q)$ is called the maximum mean discrepancy (MMD), which has been attracting much attention due to its solid theoretical foundations [SFG+09, GBR+12, GSS+12, ZGB13, DLL+18]. It is natural that MMD was introduced into GAN-type learning, named MMD-GAN [LSZ15, DRG15, STS+17, LCC+17, BSAG18, ASBG18]. The second category is the $\zeta$-divergence [CS04], which is defined as

$$\delta_\zeta(p,q) = \int_{\mathcal{X}} q(x)\zeta\left(\frac{p(x)}{q(x)}\right)dx,$$

where $\zeta$ is a convex, lower-semicontinuous function satisfying $\zeta(1) = 0$. For different $\zeta$, we have different $\delta_\zeta(p,q)$ and hence we can design different GAN models [NCT16]. For example, the pioneering GAN [GPAM+14] is based on the Jensen-Shannon divergence, and the least squares GAN is related to the Pearson $\chi^2$ divergence [MLX+17], which are $\zeta$-divergences.

In this paper, we propose a new paradigm that casts the generative adversarial learning as a goodness-of-fit (GOF) test problem. It is fundamentally different from the existing GAN models principled on two-sample tests. The aim of the goodness-of-fit (GOF) test is to determine how well a given model distribution $p$ fits a set of given samples $\mathcal{D}_x = \{x_i\}_{i=1}^n$ from an unknown distribution $q$. The knowledge of $p$ is what distinguishes the GOF test from the two-sample test, and brings higher power (i.e., probability of correctly rejecting the null hypothesis) to the GOF test statistics compared to the two-sample test statistics. Higher power in hypothesis testing suggests higher discriminability in GAN training. Specifically, by adopting the energy model to simulate the underlying distribution of the real data, we propose a deep energy adversarial network (DEAN) that casts the adversarial learning as an optimization of a GOF test statistic. We adopt a variant of finite set Stein discrepancy (vFSSD) [JXS+17] as the GOF test statistic, which is a linear time nonparametric kernel test statistic and shows stronger power than the two-sample test statistic, MMD. The proposed DEAN can be interpreted as a novel two generator game via GOF tests: One explicit generator is designed to learn an energy-based distribution (EBD), which maps the real data to a scalar energy-based probability, and the other implicit generator is trained by minimizing the vFSSD between the EBD and the generated data. We design a two-level alternative optimization procedure to train the two generators, such that the explicit one provides the formulation of the distribution and the implicit one produces genuine-looking images. It is worth noting that the DEAN framework with two generators proposed in this paper is versatile and able to yield specific training algorithms for different architectures of deep neural networks.

## 2 Related Work

**Energy-Based GANs.** Energy-based models capture dependencies over random variables by defining an energy function. The energy function maps each configuration of random variables to a scalar energy value, where lower energy values are assigned to more likely configurations. In general, the exact MLE of an energy model is challenging to calculate due to the difficulty of evaluating the normalization constant and its gradient. To overcome this difficulty, deep energy models [NCKN11, XLZW16, SMSH18] and a series of energy-based GANs have been proposed, including EBGAN [ZML17], calibrated EBGAN [DAB+17], BEGAN [BSM17] and MAGAN [WCCD17]. In this paper, we adopt the energy-based models to fit the real data, and use the resulting energy-based distribution as the known distribution in the GOF test to optimize the implicit generator.

**Score Matching and Stein's Method.** Score matching was developed for the parameter estimation of unnormalized densities caused by the partition function being eliminated in the score function

[Hyv05, SMSH18]. For the GOF test, traditional methods need to calculate the likelihoods of the models. However, for large deep generative models, this is computationally intractable due to the complexity of the models. Recently, Stein's method [S$^+$72, OGC17] was introduced into the kernel domain [GM17, WL16, LW16, LW18, FWL17, DLL$^+$19b], which combines Stein's identity with the RKHS theory. This is a *likelihood-free* method that depends on the known distribution $p$ only through logarithmic derivatives, and is closely related to score matching. The proposed statistic is referred to as kernel Stein discrepancy (KSD) [CSG16, LLJ16]. To improve the performance of KSD and MMD, Jitkrittum et al. [JXS$^+$17] proposed the finite set Stein discrepancy (FSSD) by introducing a witness function on a finite set. Inspired by [JXS$^+$17], we introduce score matching and Stein's method into the domain of generative adversarial learning, making the GOF test possible by providing one of the distributions $p$ and $q$. We eliminate the partition function by taking logarithmic derivatives of the energy-based distribution that is directly involved in optimizing the implicit generator.

**Goodness-of-fit Test for Generative Model Learning.** In recent years, there are two emerging families for generative model learning [HYSX18], generative adversarial networks (GANs) and autoencoders (AEs) or variational AEs (VAEs), which are two distinct paradigms and have both received extensive studies. Our paper and [PDB18] both introduce GOF tests into deep generative modeling, but fall into different paradigms: [PDB18] is an AE-based method without adversarial learning while our paper is a GAN-type approach. The HTAE (hypothesis testing AE) in [PDB18] minimized the reconstruction error, but no adversarial learning (min-max adversarial optimization) was involved. The statistic in this paper is a kernel-based *nonparametric* GOF statistic. The Shapiro-Wilk test in [PDB18] is a traditional *parametric* GOF statistic for testing normality.

## 3  Background

The paradigm of generative adversarial networks (GANs) [GPAM$^+$14] generates samples using a training procedure that pits a generator $G$ against a discriminator $D$. $D$ is trained to distinguish training samples from the samples produced by $G$, while $G$ is trained to increase the probability of its samples being incorrectly classified as real data. In the original formulation [GPAM$^+$14], the training procedure defines a minimax game

$$\min_G \max_D \mathbb{E}_{x \sim p(x)} \left[ \log D(x) \right] + \mathbb{E}_{z \sim p_z(z)} \left[ \log(1 - D(G(z))) \right],$$

where $p(x)$ is a data distribution in $\mathbb{R}^d$, $D$ is a function that maps $\mathbb{R}^d$ to $[0, 1]$, and $G$ is a function that maps a noise vector $z \in \mathbb{R}^m$, drawn from a simple distribution $p_z(z)$, to the ambient space of the training data. The idealized algorithm can be shown to converge and to minimize the Jensen-Shannon divergence between the data generating distribution and the distribution parameterized by $G$.

Let $\mathcal{H}_\kappa$ be a reproducing kernel Hilbert space (RKHS) defined on the data domain $\mathcal{X}$ with the reproducing kernel $\kappa : \mathcal{X} \times \mathcal{X} \to \mathbb{R}$. We consider the function class $\mathcal{F}$ as a unit ball in a universal RKHS $\mathcal{H}_\kappa$, since this class is rich enough to show equivalence between the zero expectation of the statistics and the equality of two distributions [FBJ04, SGF$^+$10, Ste01, MXZ06]. Universality requires that $\kappa$ is continuous and $\mathcal{H}_\kappa$ is dense in the space of bounded continuous functions $C(\mathcal{X})$ with respect to the $L_\infty$ norm. Gaussian and Laplacian RKHSs are universal [Ste01].

The mean embedding of a distribution $p$ in $\mathcal{F}$, written as $\mu_\kappa(p) \in \mathcal{F}$, is defined such that $\mathbb{E}_{x \sim p} f(x) = \langle f, \mu_\kappa(p) \rangle$ for all $f \in \mathcal{F}$. The squared MMD between two distributions $p$ and $q$ is the squared RKHS distance between their respective mean embeddings,

$$\mathrm{MMD}^2[\mathcal{F}, p, q] = \|\mu_\kappa(p) - \mu_\kappa(q)\|_\mathcal{F}^2 = \mathbb{E}_{zz'} h(z, z'),$$

where $z = (x, y)$, $z' = (x', y')$ and $h(z, z') = \kappa(x, x') + \kappa(y, y') - \kappa(x, y') - \kappa(x', y)$. It has been proved that for a unit ball $\mathcal{F}$ in a universal RKHS, $\mathrm{MMD}[\mathcal{F}, p, q] = 0$ if and only if $p = q$ [GBR$^+$12].

For two sets of samples $\mathcal{D}_x = \{x_i\}_{i=1}^n \subset \mathcal{X} \subseteq \mathbb{R}^d$, where $x_i \sim p$ i.i.d., and $\mathcal{D}_y = \{y_j\}_{j=1}^m \subset \mathcal{Y} \subseteq \mathbb{R}^d$, where $y_j \sim q$ i.i.d., if we assume $m = n$, the minimum variance unbiased estimator of $\mathrm{MMD}^2[\mathcal{F}, p, q]$ can be represented as

$$\mathrm{MMD}_{\mathrm{Unb}}^2[\mathcal{F}, p, q] = \frac{1}{n(n-1)} \sum_{i \neq j} h(z_i, z_j).$$

The typical two-sample test based GANs are MMD GANs [LSZ15, DRG15, STS$^+$17, LCC$^+$17] which train the parameter $\theta_g$ by optimizing

$$\underset{\theta_g}{\arg\min} \, \mathrm{MMD}^2_{\mathrm{Unb}}[\mathcal{F}, p, G(z; \theta_g)].$$

# 4 Deep Energy Adversarial Network

In this section, we present the deep energy adversarial network (DEAN). Our primary contribution is a new paradigm for generative adversarial learning, which consists of two generative networks: an explicit one that learns an energy-based distribution (EBD) fitting the real data, and an implicit one that produces genuine-looking images by minimizing the discrepancy between the underlying distribution of the real data and the EBD produced by the explicit generative network.

The following characteristics make the proposed DEAN distinguishable from the existing GANs. First, DEAN makes it possible for the generative adversarial learning to approximate the underlying distribution of the real data, not just produce fake data that mimics the real data. Second, the GOF test is adopted to replace the two-sample test, such that the knowledge of $p$ is used to increase the test power (probability of correctly rejecting the null hypothesis). This power can be understood as the discriminability in GAN training. Third, DEAN can be considered an algorithm for training deep energy-based models [NCKN11, XLZW16, SMSH18], where the implicit generator is used to provide "negative" samples. Fourth, the explicit generator plays the role similar to the discriminator of the existing GAN models. DEAN is a two generator game.

## 4.1 Energy Estimator Network

Energy-based models $\mathcal{E}_\theta(x) : \mathcal{X} \to \mathbb{R}$ associate an energy value with a sample $x$, where $\theta$ are the parameters. Ideally, high energy is assigned to the generated fake data, and low energy to real data. We can obtain a distribution based on $\mathcal{E}_\theta(x)$,

$$p(x; \theta) = \frac{1}{Z_\theta} \exp(-\mathcal{E}_\theta(x)).$$

The parameters $\theta$ of the energy function are often learned to maximize the likelihood of the data; the main challenge in this optimization is evaluating the partition function $Z_\theta = \int_x \exp(-\mathcal{E}_\theta(x))$, which is an intractable sum or integral for most high-dimensional problems.

Now we define the loss function of the explicit generative network (EGN) of DEAN as follows:

$$\min_{\theta_e} \mathcal{E}(x; \theta_e) + \left[ \gamma - \mathcal{E}\left( G(z; \theta_g^*); \theta_e \right) \right]^+, \tag{1}$$

where $\mathcal{E}(x; \theta_e)$ is an *energy model* parameterized by $\theta_e$, $[\cdot]^+ = \max(\cdot, 0)$ and $\gamma$ is a given positive margin. We can use $\mathcal{D}_{x'} = \{x_i' := G(z_i; \theta_g^*)\}_{i=1}^n$ to denote the generated fake samples with $\theta_g^*$ optimized in the implicit generator network, where $n$ is the batch size. $\mathcal{D}_{x'}$ and the real data $\mathcal{D}_x = \{x_i\}_{i=1}^n$ are both fed into Equation (1), where the real data $\mathcal{D}_x = \{x_i\}_{i=1}^n$ is forced to have low energy, while generated fake data $\mathcal{D}_{x'}$ is forced to have high energy. This loss function (1) is possibly the simplest energy-based loss and is the same as that of EBGAN [ZML17]. However, for the DEAN framework, other energy-based losses can also be adopted, such as the losses in calibrated EBGAN [DAB$^+$17], BEGAN [BSM17] and MAGAN [WCCD17].

When the network parameters $\theta_e^*$ are optimized, we can define a probability distribution

$$p(x; \theta_e^*) = \frac{1}{Z_{\theta_e^*}} \exp(-\mathcal{E}(x; \theta_e^*)).$$

We take two cases of $\mathcal{E}(x; \theta_e)$ as examples. First, we consider the Gaussian-Bernoulli restricted Boltzmann machine (RBM) [HS06], which is a hidden variable graphical model consisting of a continuous observable variable, $x \in \mathbb{R}^d$, and a binary hidden variable, $r \in \{\pm 1\}^{d_h}$ [1]. We write

$$\mathcal{E}(x; \theta_e) = \frac{1}{2} \|x\|^2 - b^{\mathrm{T}} x - \varsigma(B^{\mathrm{T}} x + c),$$

where $\theta_e = \{b, B, c\}$ and $\varsigma(v) = \sum_{i=1}^{n} \log(\exp(v_i) + \exp(-v_i))$. For optimized $\theta_e^*$, we have

$$p(x; \theta_e^*) = \frac{1}{Z_{\theta_e^*}} \exp(-\mathcal{E}(x; \theta_e^*)).$$

Second, we consider a deep auto-encoder as a more complex energy model

$$\mathcal{E}(x; \theta) = \|x - \mathrm{AE}(x; \theta_e)\|,$$

where $\mathrm{AE}(x; \theta_e)$ denotes a deep auto-encoder parameterized by $\theta_e$. For the optimized parameters $\theta_e^*$, we can define

$$p(x; \theta_e^*) = \frac{1}{Z_{\theta_e^*}} \exp(-\mathcal{E}(x; \theta_e^*)).$$

In the implicit generator network of DEAN shown in the next section, we will introduce a score function [Hyv05] to avoid calculating the partition function $Z_\theta$,

$$s(x, \theta) = \nabla_x \log p(x, \theta) = -\nabla_x \mathcal{E}(x, \theta_e^*),$$

since $Z_\theta$ is independent of $x$. We will fully exploit the knowledge of the distribution $p$ by introducing the Stein operator [S$^+$72, OGC17]. It is the knowledge of $p$ that distinguishes the GOF test from the two-sample test, and makes the DEAN paradigm fundamentally different from the existing GANs.

## 4.2 GOF-driven Generator Network

We present the implicit generative network (IGN) of DEAN, which is trained by minimizing a GOF test statistic between the energy-based distribution $p(x; \theta_e^*)$ learned by the EGN and the generated (fake) data, such that the underlying distribution of the generated data is close to $p(x; \theta_e^*)$.

We first introduce the Stein operator [S$^+$72, OGC17], which depends on the distribution $p$ only through logarithmic derivatives. A Stein operator $T_p$ takes a multivariate function $f(x) = (f_1(x), \ldots, f_d(x))^{\mathrm{T}} \in \mathbb{R}^d$ as input and outputs a function $(T_p f)(x) : \mathbb{R}^d \to \mathbb{R}$. The function $T_p f$ has the key property that, for all $f$s in an appropriate function class, $\mathbb{E}_{x \sim q}[(T_p f)(x)] = 0$ if and only if $p = q$. Thus, this expectation can be used to test the goodness-of-fit: how well a model distribution $p$ fits a given set of samples $\{x_i\}_{i=1}^{n} \subset \mathcal{X} \subseteq \mathbb{R}^d$ from an unknown distribution $q$.

We consider the function class $\mathcal{F}^d := \mathcal{F} \times \cdots \times \mathcal{F}$, where $\mathcal{F}$ is a unit-norm ball in a universal RKHS. Assume that $f_i \in \mathcal{F}$ for all $i = 1, \ldots, d$ such that $f \in \mathcal{F}^d$ with the inner product $\langle f, g \rangle_{\mathcal{F}^d} := \sum_{i=1}^{d} \langle f_i, g_i \rangle_{\mathcal{F}}$ for $g \in \mathcal{F}^d$. According to the reproducing property of $\mathcal{F}$, $f_i(x) = \langle f_i, \kappa(x, \cdot) \rangle_{\mathcal{F}}$, and $\frac{\partial \kappa(x, \cdot)}{\partial x_i} \in \mathcal{F}$, we define $\omega_p(x, \cdot) = \frac{\partial \log p(x)}{\partial x} \kappa(x, \cdot) + \frac{\kappa(x, \cdot)}{\partial x}$. Kernel Stein operator can be written as

$$(T_p f)(x) = \sum_{i=1}^{d} \left( \frac{\partial \log p(x)}{\partial x_i} f_i(x) + \frac{\partial f_i(x)}{\partial x_i} \right) = \langle f, \omega_p(x, \cdot) \rangle_{\mathcal{F}^d}.$$

Now we introduce the kernel Stein discrepancy (KSD) [CSG16, LLJ16], which is formulated as

$$\mathrm{KSD}[\mathcal{F}^d, p, \mathcal{D}_x] = \sup_{\|f\|_{\mathcal{F}^d} \leq 1} \langle f, \mathbb{E}_{x \sim q} \omega_p(x, \cdot) \rangle := \|g(\cdot)\|_{\mathcal{F}^d}, \tag{2}$$

where $g(\cdot) = \mathbb{E}_{x \sim q} \omega_p(x, \cdot)$.

Let $V = \{v_1, \ldots, v_J\} \subset \mathbb{R}^d$ be random vectors drawn from a distribution, where $J$ is a pre-defined hyper-parameter. The statistic of the finite set Stein discrepancy (FSSD) [JXS$^+$17] is defined as

$$\mathrm{FSSD}[\mathcal{F}^d, p, \mathcal{D}_x] = \frac{1}{dJ} \sum_{i=1}^{d} \sum_{j=1}^{J} g_i^2(v_j),$$

where $g(\cdot)$ is referred to as the Stein witness function, given in Equation (2).

In the following, we present a variant of FSSD as the loss function of the IGN of DEAN. Let $\Omega(x) \in \mathbb{R}^{d \times J}$, such that

$$\Omega(x)_{i,j} = \omega_{p,i}(x, v_j)/\sqrt{dJ}, \quad \tau(x) = \mathrm{vec}(\Omega(x)) \in \mathbb{R}^{dJ},$$

where $\mathrm{vec}(\cdot)$ denotes the vectorization of matrices. The unbiased estimator of FSSD is defined as

$$\widehat{\mathrm{FSSD}}^2[\mathcal{F}^d, p, \mathcal{D}_x] = \frac{2}{n(n-1)} \sum_{i<j} \Delta(x_i, x_j),$$

where $\Delta(x, y) = \tau(x)^{\mathrm{T}} \tau(y)$. Without loss of generality, we will adopt $\widehat{\mathrm{FSSD}}^2[p, \mathcal{D}_x]$ as an abbreviation, since the function class $\mathcal{F}^d$ is fixed when the kernel is given.

Now we present a variant of $\widehat{\mathrm{FSSD}}^2[p, \mathcal{D}_x]$ as the loss function. The reason for introducing a variant is as follows. According to Proposition 2 in [JXS+17], under the alternative hypothesis $H_1 : p \neq q$,

$$n\widehat{\mathrm{FSSD}}^2 \sim \sqrt{n}\mathcal{N}(0, \sigma_{H_1}) + n\mathrm{FSSD}^2,$$

if $\sigma_{H_1} = 4\mu^{\mathrm{T}}\Sigma_q\mu > 0$, where $\mu = \mathbf{E}_{x \sim q}[\tau(x)]$ and $\Sigma_q = \mathrm{cov}_{x \sim q}[\tau(x)] \in \mathbb{R}^{dJ \times dJ}$. From the above equation, we know that $n\widehat{\mathrm{FSSD}}^2$ is highly dependent on the dimension of the data: when the dimension $d$ increases, the dimension of $\Sigma_q$ will also increase, and then the variance $\sigma_{H_1}$ becomes larger. When the variance becomes larger, the resulting values of the statistic will become unstable. To alleviate the impact of dimension and stabilize the statistic, we introduce

$$\mathrm{vFSSD}[p, \mathcal{D}_x] = \frac{1}{\hat{\sigma}_{H_1}} \widehat{\mathrm{FSSD}}^2[p, \mathcal{D}_x]$$

as the variant of $\widehat{\mathrm{FSSD}}^2[p, \mathcal{D}_x]$ [JXS+17], where $\hat{\sigma}_{H_1}$ is an empirical estimate of $\sigma_{H_1} = 4\mu^{\mathrm{T}}\Sigma_q\mu$, which is the empirical variance of the limiting distribution of $\sqrt{n}\left(\widehat{\mathrm{FSSD}}^2 - \mathrm{FSSD}^2\right)$. Now we define the loss function of the IGN as follows:

$$\min_{\theta_g} \max_{\xi} \mathrm{vFSSD}_{\xi}\left[p(x; \theta_e^*), \mathcal{D}_{x'}\right], \tag{3}$$

where $\xi = \left\{ \{v_i\}_{i=1}^J, \sigma_k \right\}$ denotes the hyper-parameters of vFSSD, including the kernel parameter $\sigma_k$ and $J$ test locations $\{v_i\}_{i=1}^J$.

**Remark:** In Equation (3), the inner maximum is used to optimize the hyper-parameters $\xi$ of IGN itself. This is similar to the idea of Equation (3) in [LCC+17]. We can set random values for the hyper-parameters $\xi$. If so, we solve the DEAN framework by alternately optimizing the loss function of EGN (Equation (1)) and the loss function of IGN (Equation (3)) with fixed $\xi$. However, maximizing Equation (3) with respect to the hyper-parameters $\xi$ can increase the test power of vFSSD[2], which will eventually force the IGN to produce more realistic-looking images.

Therefore, we present the following two objectives, (4) and (5), to optimize Equation (3) and improve the test power of DEAN.

$$\max_{\xi} \mathrm{vFSSD}_{\xi}\left[p(y; \theta_e^*), \mathcal{D}_{x'*}\right], \tag{4}$$

where $\mathcal{D}_{x'*} = \left\{ x_i'^* := G(z_i; \theta_g^*) \right\}_{i=1}^n$ and $G(z_i; \theta_g^*)$ is a deep network with the optimized parameter $\theta_g^*$. The hyper-parameters $\xi = \left\{ \{v_i\}_{i=1}^J, \sigma_k \right\}$ will be optimized in Equation (4).

$$\min_{\theta_g} \mathrm{vFSSD}_{\xi^*}\left[p(y; \theta_e^*), \mathcal{D}_{x'}\right], \tag{5}$$

where $\xi^* = \left\{ \{v_i^*\}_{i=1}^J, \sigma_k^* \right\}$ denotes the optimized hyper-parameters, and the parameters $\theta_g$ for $\mathcal{D}_{x'} = \{x_i' := G(z_i; \theta_g)\}_{i=1}^n$ will be optimized.

In summary, DEAN is solved by alternately optimizing Equation (1) and Equation (3); Equation (3) is solved by alternatively optimizing Equation (4) and Equation (5), if necessary. It is a two-level alternative optimization procedure. The energy-based probability $p(x; \theta_{e^*})$, playing the role of a discriminator, is trained to provide low energy to the real data, and high energy to the fake data produced by the IGN $G(z, \theta_g)$. The IGN is trained by minimizing vFSSD between the generated data

and $p(x; \theta_e^*)$, such that the underlying distribution of the generated data gradually becomes closer to $p(x; \theta_e^*)$ that fits the real data.

Finally, we characterize the solutions of DEAN. Let $p_x$ and $p_{x'}$ be the distributions of real and fake data; $p_e$ denotes the energy-based distribution. In DEAN, $p_e$ is a bridge connecting $p_x$ and $p_{x'}$. For the IGN, the network is trained to have $p_{x'}$ equal to $p_e$. Please refer to Theorem 1, which can be easily proved based on Theorem 1 of [JXS+17]. For the EGN, $p_e$ is learned to estimate $p_x$. Please see Theorem 2, which can be proved according to Theorem 1 of [ZML17] and Theorem 1 of [GPAM+14] with fixed $p_{x'}$. Different from GANs, which are implicit generative models (IGMs), DEAN can explicitly estimate the underlying distribution of the real data after estimating $\theta_e$ and $\theta_g$.

**Theorem 1** *We assume that $\mathcal{D}_{x'}$ is drawn from $p_{x'}$. If $\kappa$ is a universal and analytic kernel;* $\mathbf{E}_{a \sim p_{x'}} \mathbf{E}_{b \sim p_e} \left[ s^{\mathrm{T}}(a)s(b)\kappa(a,b) + s^{\mathrm{T}}(b)\nabla_a\kappa(a,b) + s^{\mathrm{T}}(a)\nabla_b\kappa(a,b) + \sum_{i=1}^{d} \frac{\partial^2 \kappa(a,b)}{\partial a_i \partial b_i} \right] < \infty$ *with $s(a) = \nabla_a \log p_e(a)$; $\mathbf{E}_{a \sim p_{x'}} \|\nabla_a \log p_e(a) - \nabla_a \log p_{x'}(a)\|^2 < \infty$; $\lim_{\|a\| \to \infty} p_e(a)g(a) = 0$, where $g(\cdot)$ is given in Eq. (2) in Section 4.2; for any $J \geq 1$, almost surely $\mathrm{FSSD}[p_e, \mathcal{D}_{x'}] = 0$ if and only if $p_{x'} = p_e$.*

**Theorem 2** *Let $\Lambda(\theta_e) = \mathcal{E}(x; \theta_e) + \left[\gamma - \mathcal{E}\left(G(z; \theta_g^*); \theta_e\right)\right]^+$. The minimum of $\Lambda(\theta_e)$ is achieved if and only if $p_e = p_x$. With the optimized $\theta_e^*$, $\int_{x,z} \Lambda(\theta_e^*)p_x(x)p_z(z)\mathrm{d}x\mathrm{d}z = \gamma$.*

## 5  Experiments

Here, we conduct experiments to evaluate the performance of the proposed DEAN as compared with the existing GAN models.

We compared five related GAN models, DCGAN [RMC16], EBGAN [MLS+17], WGAN-GP [SGZ+16], MMD-GAN [LCC+17, BSAG18] and Scaled MMD-GAN (SMMD-GAN) [ASBG18].

The evaluations are conducted on three popular datasets, including MNIST [LBBH98] (70,000 images, $28 \times 28$), CIFAR-10 [KH09] (60,000 images, $32 \times 32$), and CelebA [YLLT15] (202,599 face images, resized and cropped to $160 \times 160$).

For MNIST and CIFAR-10, the IGN of DEAN adopts a DCGAN generator [RMC16] with vFSSD as the loss function. An auto-encoder with convolutional layers is adopted as the EGN of DEAN (analogous to the discriminators of the existing GANs). The loss of the discriminator is defined in (1), where we set $\gamma = 1$ as in [ZML17], and $\mathcal{E}(x; \theta) = \|x - \mathrm{AE}(x; \theta_e)\|^3$. For CelebA, we use a ResNet as the IGN and an auto-encoder as the EGN. The input noise vector $z \in \mathbb{R}^{128}$ for the generator (IGN) is independently drawn from a standard normal distribution.

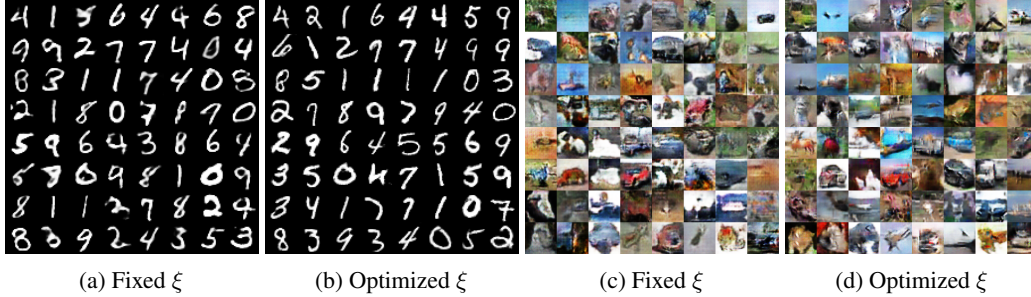

<div align="center">

(a) Fixed $\xi$       (b) Optimized $\xi$       (c) Fixed $\xi$       (d) Optimized $\xi$

</div>

Figure 1: Images generated by DEAN with fixed and optimized hyper-parameters $\xi$ on MNIST and CIFAR-10.

Kernel selection is import to the performance of kernel methods [DL14b, DL14a, DL17, LLD+18, LLD+18, DLL+19a, LLJ+19]. We introduce the mixture of linear and rational quadratic functions given in [BSAG18] as kernel functions for the DEAN framework: $\kappa^{dot+rq}(x,y) = \kappa^{dot}(x,y) + \kappa^{rq}(x,y)$, $\kappa^{dot} = \langle x, y \rangle$, $\kappa^{rq}(x,y) = \sum_{\alpha \in \mathcal{A}} \kappa_\alpha^{rq}(x,y)$, $\kappa_\alpha^{rq}(x,y) = \left(1 + \frac{\|x-y\|^2}{2\alpha}\right)^{-\alpha}$, where

$\mathcal{A} = \{0.2, 0.5, 1, 2, 5\}$. If we simply calculate pixel-level kernels, the performance is poor due to the high dimension of the images. Following [LCC$^+$17], we consider kernels defined on top of a low-dimensional representation $\phi_\theta : \mathcal{X} \rightarrow \mathbb{R}^s$, which implies that $\kappa_\theta^{dot+rq}(x, y) = \kappa^{dot+rq}(\phi_\theta(x), \phi_\theta(y))$. In DEAN, we adopt the output of the inner layer of the auto-encoder as the low-dimensional representation $\phi_\theta$. We set the number of test locations $J = 5$ to compute the value of vFSSD.

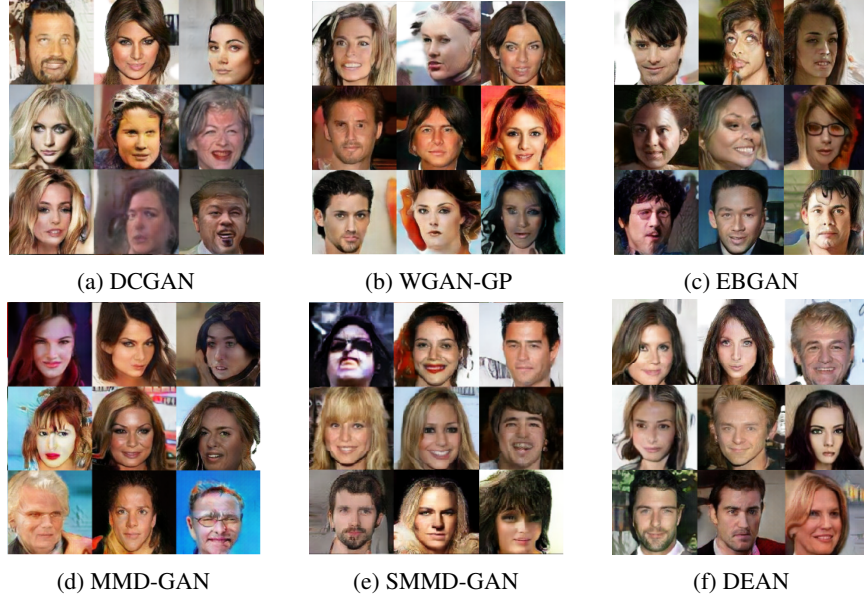

| | | |
|---|---|---|
| (a) DCGAN | (b) WGAN-GP | (c) EBGAN |
| (d) MMD-GAN | (e) SMMD-GAN | (f) DEAN |

Figure 2: Faces generated by different GAN models trained on CelebA.

All models are trained on an NVIDIA Tesla V100 GPU. We adopt five EGN updates per IGN step. For IGN, we optimize the hyper-parameter $\xi$ by Equation (4) for every update. We use initial learning rates of 0.0001 for MNIST, CIFAR-10 and CelebA. We use the Adam optimizer [KB15] with $\beta_1 = 0.5$, $\beta_2 = 0.9$.

We show the images generated by DEAN trained on MNIST and CIFAR-10 in Figure 1. We have two observations: a) DEAN can produce genuine-looking images; b) the quality of the images generated with optimized hyper-parameter $\xi$ is better than that with fixed hyper-parameter. The generated images trained on CelebA are shown in Figure 2. We can find that faces generated by DEAN are realistic-looking. In the compared methods, the face quality of SMMD-GAN is better than that of other GANs. However, the existing MMD loss may discourage the learning of fine details in data [WSH19]. Therefore, higher discriminability of the loss function and automatically tunable hyper-parameters of DEAN may help to learn fine details of images.

## 6 Conclusions

In this paper, we established the connection between the goodness-of-fit (GOF) test and generative adversarial learning, and proposed a new adversarial learning paradigm. It is a game between two generative networks, fundamentally different from the existing GANs principled on two-sample tests, which may open the door for research into generative-to-generative adversarial learning models. Empirical evaluations have shown that DEAN can achieve high quality generations as compared to the state-of-the-art approaches. Besides GOF-test GANs, in the near future, we will study independence-test GANs via Hilbert Schmidt independence criterion (HSIC) [GFT$^+$08].

**Acknowledgments**

This work was supported in part by National Natural Science Foundation of China (No. 61703396), the CCF-Tencent Open Fund and Shenzhen Government (GJHZ20180419190732022).

## Footnotes

[1] The joint probability distribution of $x$ and $r$ is $p(x, r) = \frac{1}{Z_{\theta_e}} \exp(x^{\mathrm{T}} B r + b^{\mathrm{T}} x + c^{\mathrm{T}} x - \frac{1}{2} \|x\|^2)$.

[2]Please refer to Proposition 4 of [JXS+17].

[3]We also adopted RBM as the energy function at the initial stage. However, the performance of DEAN with RBM is not comparable to that with autoencoder, so we discarded the results.

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
