[Reviews · NeurIPS 2019]

Reviewer 1



====== After reviewing the author feedback, my updated comments are as follows: The theorems are a good addition and clear things up; I recommend including them. Regarding the experiment comparing MMD, Lin-MMD, and FSSD: I don’t doubt that FSSD yields a more powerful test than MMD (as you say, a GOF test has access to information which isn’t available in a two-sample test). But GANs (even MMD-GANs) don’t just minimize the MMD; they minimize a loss involving a discriminator which was estimated from potentially millions of samples in past iterations. Similarly, DEAN doesn’t just minimize the FSSD; it minimizes an estimate of the FSSD where the true energy is replaced by a model which was estimated from samples (as with GANs, potentially millions of samples). Overall I raise my score to a 6, with the strong recommendation that the authors back away from justifying DEAN through arguments about GOF tests being more efficient than two-sample tests. GANs are not two-sample tests, at least not between minibatches. It's a novel GAN algorithm, and it gives you an energy model "for free"; that should be justification enough. ======== The idea of training the generator in a GAN by using a GOF test is quite novel (to my knowledge) and interesting (in my opinion). GANs have computational and statistical challenges, and exploring the space of possible formulations has the potential to lead to improvements -- I consider this paper a good exploration of a formulation which is quite different from the usual ones, even if it doesn't result in any improvements. The paper is generally written clearly; the derivation and the experiments are easy enough to follow. This paper seems to be missing a characterization of the solutions. In particular, is DEAN consistent (i.e. at equilibrium, the IGN recovers the data density) for any choice of energy model objective? L158-161 seems to suggest so, but it doesn't seem intuitive to me. GOF tests are often more powerful than two-sample tests, and this paper seems to imply that this gives DEAN improved statistical power over usual GAN techniques (e.g. L61-65, L170-171, L175-176); in fact it seems to be the main selling point of the method. However, this claim isn't formally stated or explained in detail, nor backed by any theorem or experiment in the paper. I'd like to see the claim be made clearer and more explicit, and also a demonstration of this improved statistical power in a toy setting where the exact solution is known and exact optimization is possible. If the claim is improved statistical power, I'd want to see a plot showing faster convergence on a toy problem in terms of sample size. Regarding the experiments: I think they're sufficient for the claim that "DEAN performs roughly on par with a good GAN", but not to demonstrate any practical improvement -- for that you'd either need a bigger effect size, or a much more careful setup (e.g. human eval against strong baselines). I like the novelty of the idea behind this paper and would very much like to see it accepted, but in its current state I think improvements are needed; hence I vote to reject.

Reviewer 2



The proposed model has two networks - one network to explicitly learn the energy distribution, and a second implicit generation network which is trained by minimizing goodness of fit statistic between the energy distribution and the generated data. The authors demonstrate how Stein operator can be used for training the implicit generator, resulting in a two step training procedure. Originality: The proposed approach of minimizing the gof statistic between generated data and an energy distribution seems novel. The authors have clearly highlighted their approach in context of prior works on energy-based GANs. Clarity: The paper is well-written, though some more discussion can be added in the experiments on some additional takeaways and insights. It's not clear if authors considered two different energy functions, one autoencoder based and another RBM based? A comparison of the two doesn't come up in the experiments. T Quality: Authors experiment with 3 datasets, and compare against 5 baseline method. They report Inception Score, Fréchet Inception Distance and Kernel Inception Distance. The examples in Figure 2 generated from the proposed DEAN model seem to be of high quality. It would have been great to see some crowdsourced experiments to judge the quality of generated images. Additionally, there aren't many insights about the method apart from better quantitative measures.

Reviewer 3



1) This paper proposes a new paradigm that casts the generative adversarial learning as a goodness-of-fit (GOF) test problem. However, the following paper (published in UAI 2018) already proposed this paradigm. http://auai.org/uai2018/proceedings/papers/356.pdf It is not clear what new contribution this NeurIPS 2019 paper has made, compared to the UAI 2018. BTW, this paper does not cite the UAI 2018 paper. For the above reason, the originality of the proposed GOF paradigm is in doubt. 2) The quality of the paper is good. 3) The proposed technique is clearly described.

[Author Response · NeurIPS 2019]

We would like to thank all three reviewers for acknowledging our contributions and providing valuable feedback. Please
find our responses to your comments below.

**Reviewer #1:**
Thank you for the positive comments on the novelty of our idea and insightful questions for further improvement.
We first characterize the solutions of DEAN. Let $p_x$ and $p_{x'}$ be the distributions of real and fake data; $p_e$ denotes the
energy-based distribution. In DEAN, $p_e$ is a bridge connecting $p_x$ and $p_{x'}$. Now we provide two theorems for the
characterization. For the IGN, the network is trained to have $p_{x'}$ equal to $p_e$. Please refer to Theorem 1, which is proved
based on Theorem 1 of [JXS$^+$17]. For the EGN, $p_e$ is learned to estimate $p_x$. Please see Theorem 2, which is proved
according to Theorem 1 of [ZML17] and Theorem 1 of [GPAM$^+$14]. At present, Theorem 2 is proved with $\Lambda(\theta_e)$.
Other choices for the energy objective will be left to future works. Detailed proofs of the following theorems will
be given in the Supplement of the final version. Different from GANs, which are implicit generative models (IGMs),
DEAN can explicitly estimate the underlying distribution of the real data after estimating $\theta_e$ and $\theta_g$.

**Theorem 1** *We assume that $\mathcal{D}_{x'}$ is drawn from $p_{x'}$. If the following conditions are satisfied: $\kappa$ is a universal and*
*analytic kernel; $\mathbf{E}_{a \sim p_x}\mathbf{E}_{b \sim p_e}\left[s^{\mathrm{T}}(a)s(b)\kappa(a,b) + s^{\mathrm{T}}(b)\nabla_a\kappa(a,b) + s^{\mathrm{T}}(a)\nabla_b\kappa(a,b) + \sum_{i=1}^{d}\frac{\partial^2\kappa(a,b)}{\partial a_i\partial b_i}\right] < \infty$ with*
$s(a) = \nabla_a\log p_e(a)$; $\mathbf{E}_{a \sim p_{x'}}\|\nabla_a\log p_e(a) - \nabla_a\log p_{x'}(a)\|^2 < \infty$; $\lim_{\|a\|\to\infty}p_e(a)g(a) = 0$, where $g(\cdot)$ is given
*in Eq. (2) in Section 4.2; for any $J \geq 1$, almost surely $\mathrm{FSSD}[p_e, \mathcal{D}_{x'}] = 0$ if and only if $p_{x'} = p_e$.*

**Theorem 2** *Let $\Lambda(\theta_e) = \mathcal{E}(x;\theta_e) + \left[\gamma - \mathcal{E}\left(G(z;\theta_g^*);\theta_e\right)\right]^+$ (please refer to Eq. (1) in Section 4.1 for details). The*
*minimum of $\Lambda(\theta_e)$ is achieved if and only if $p_e = p_x$. With the optimized $\theta_e^*$, $\int_{x,z}\Lambda(\theta_e^*)p_x(x)p_z(z)\mathrm{d}x\mathrm{d}z = \gamma$.*

Following your suggestion, we compare the powers (successful rejection rates) of MMD, linear-time MMD [GBR$^+$12]
and FSSD on toy problems, where MMD is a two-sample test statistic and FSSD is used for the GOF test.
We adopt the distributions Gaussian $p(x) = \mathcal{N}(x|0, \mathbf{I}_d)$ and
Laplacian $q(x) = \prod_{i=1}^{d}\mathrm{Laplace}(x_i|0, 1/\sqrt{2})$ for $d = 1, 3$, in
which the parameters are set to make $p$ and $q$ have the same mean
and variance so that the difference between $p$ and $q$ is subtle. F-
SSD shows a higher power to discriminate the subtle difference
(Figure 1). For larger sample sizes, the power of MMD is close to
that of FSSD. However, in the GAN-type training, the batch size
is usually less than 512. As the adversarial training continues, the
distribution of the generated data gets closer to the energy-based
distribution, and hence the difference becomes subtle. At this
time, the power (discriminability) of FSSD for the subtle differ-

Figure 1: Rejection rates for $d = 1$ (left) and $d = 3$.

ence becomes important for generating high-quality images. Hopefully we have cleared up your main concerns with
these theoretical and experimental discussions. We believe that the DEAN paradigm is promising, being versatile to
yield specific training algorithms for different architectures of deep networks in different domains.

**Reviewer #2:**
Thank you very much for the encouraging comments and valuable suggestions.
Following your recommendation, we will add more discussions in the experimental part to provide takeaways and
insights about DEAN. We adopted RBM as the energy function at the initial stage. However, the performance of DEAN
with RBM is not comparable to that with autoencoder, so we discarded the results. We will add clarity on this in the
final manuscript.

**Reviewer #3:**
Thank you very much for the positive comments and reasonable doubt.
In recent years, there are two emerging families for generative model learning, generative adversarial networks (GANs)
and autoencoders (AEs) or variational AEs (VAEs), which are two distinct paradigms and have both received extensive
studies. Goodness-of-fit (GOF) tests are a fundamental tool in statistical analysis, dating back to the Kolmogorov test in
1933. Our manuscript and [PDB18] both introduce GOF tests into deep generative modeling, but fall into different
paradigms: [PDB18] is an AE-based method without adversarial learning while our paper is a GAN-type approach. The
HTAE (hypothesis testing AE) in [PDB18] minimized the reconstruction error, but no adversarial learning (min-max
adversarial optimization) was involved. The statistic in our manuscript is a kernel-based *nonparametric* GOF statistic.
The Shapiro-Wilk test in [PDB18] is a traditional *parametric* GOF statistic for testing normality. Our paper is quite
different from [PDB18]. The proposed DEAN with two generators is a pioneering work in the adversarial learning
setting. Following your comment, we will cite [PDB18] in the final version.

**References**

[PDB18] Aaron Palmer, Dipak Dey, and Jinbo Bi. Reforming generative autoencoders via goodness-of-fit hypothesis testing. In
*UAI*, pages 1009–1019, 2018.


[Meta-Review · NeurIPS 2019]

Good paper introducing a new kind of generative model based on a game between an energy model and an implicit generator. Although the experimental results don't demonstrate significant gains compared to GANs, this seems like an interesting idea other researchers could follow up on.